# Sodium Butyrate Supplementation Modulates Neuroinflammatory Response Aggravated by Antibiotic Treatment in a Mouse Model of Binge-like Ethanol Drinking

**DOI:** 10.3390/ijms232415688

**Published:** 2022-12-10

**Authors:** Lei Gao, Daryl L. Davies, Liana Asatryan

**Affiliations:** Titus Department of Clinical Pharmacy, School of Pharmacy, University of Southern California, Los Angeles, CA 90033, USA

**Keywords:** gut-brain axis, ethanol-induced neuroinflammation, sodium butyrate supplementation, microglia and astrocytes, pro-inflammatory cytokines

## Abstract

Growing evidence supports the pivotal role of the bidirectional interplay between the gut microbiota and the central nervous system during the progression of alcohol use disorder (AUD). In our previous study, supplementation with sodium butyrate (SB) in C57BL/6J mice prevented increased ethanol consumption in a binge-like drinking paradigm (DID) as a result of treatment with a non-absorbable antibiotic cocktail (ABX). In this study, we tested the hypothesis that SB protection against enhanced ABX-induced ethanol consumption in mice is partially due to modulation of neuroinflammatory responses. Pro- and anti-inflammatory cytokines, as well as changes in microglia and astrocytes were analyzed in hippocampus tissues from ABX-, SB-, ABX+SB-treated mice subjected to 4-week DID. We found that ethanol without or with ABX treatment increased mRNA levels of key brain cytokines (MCP-1, TNF-α, IL-1β, IL-6 and IL-10) while SB supplementation prevented these changes. Additionally, SB supplementation prevented changes in microglia, i.e., increase in Iba-1 positive cell number and morphology, and in astrocytes, i.e., decrease in GFAP-positive cell number, induced by combination of ethanol and ABX treatments. Our results suggest that gut microbiota metabolites can influence drinking behavior by modulation of neuroinflammation, highlighting the potential for microbiome-targeting strategies for treatment or prevention of AUD.

## 1. Introduction

Alcohol use disorder (AUD) is a significant problem with over 88,000 excess deaths annually costing more than $250 billion to the American society [1,2,3,4,5]. Due to its poorly understood pathophysiology and the existence of limited FDA-approved pharmacotherapies with low rates of use and effectiveness, there is urgent need to identify novel targets for prevention and treatment of AUD. 

The neurobiology of AUD pathology is complex, involving interactions among several neurobehavioral systems [6], which are also affected from peripheral input, including the gut-brain axis. The dysregulation of the gut-brain axis has been increasingly recognized as a contributor to neurological pathologies including AUD [7]. In support, intestinal integrity is commonly damaged during chronic alcohol consumption and is known as “leaky gut” [7,8]. Alcohol-induced bacterial overgrowth and dysbiosis were found during long-term alcohol consumption in humans and in rodent models of drinking [7,9]. Moreover, changes in metabolites derived from gut microbiota have been linked to AUD-related behaviors and tissue pathologies. In this regard, reduction in commensal microbiota that produce butyrate, a short-chain fatty acid with protective properties, has been demonstrated [10]. Butyrate is a well-recognized major energy source for enterocytes [11,12], and provides protective effects to the intestinal wall [13]. Butyrate can also exert anti-inflammatory effects within the intestinal compartment [14] as well as in the nervous system [10,12] by acting as histone deacetylase inhibitor and binding to G-protein coupled receptors [15]. By virtue of its pleiotropic effects, increasing butyrate levels may be beneficial in maintaining the integrity of the gut-brain axis during chronic alcohol consumption. 

Targeting the neuroimmune system is a new avenue for developing novel treatments for neurological diseases. AUD and other substance use disorders were shown to have underlying neuroimmune mechanisms [1,16,17,18]. A few studies, investigating the causal link between changes in host microbiome and alcohol abuse behaviors, pointed to the importance of neuroinflammation [1,19]. Neuroinflammatory responses can be mediated by several key inflammatory cytokines and chemokines such as IL-1β, IL-6, TNFα, MCP-1. These inflammatory mediators have been shown to affect neurons and were associated with an array of disease symptoms in animals models (reviewed in [20]). Furthermore, increased levels of inflammatory cytokines were observed in post-mortem human brains of AUD patients and rodent brains after alcohol intake [21,22]. Although many studies focus primarily on the impact of excessive ethanol consumption on pro-inflammatory cytokines [23], several others further their research by determining the influence of anti-inflammatory cytokines such as IL-10 [24,25]. These studies indicate that pro- and anti-inflammatory cytokines are both capable of modulating the ethanol consumption behavior. 

Many of these key mediators are produced by activated resident CNS glia cells including microglia and astrocytes [26]. Microglia function as the primary immune surveillance cells of the CNS and perform macrophage-like activities [27]. Microglia also have an important impact on sensing and responding to ethanol consumption involving multiple immune signaling pathways [28]. There is a large body of work suggesting that microglia are important players contributing to ethanol-related behaviors and brain damage [29,30]. Recent research has also identified the role of astrocytes, the most numerous glial cells that support neurons, as critical regulators of neuroinflammation and alcohol consumption behaviors [31,32,33]. Ethanol-induced effects on astrocyte physiology appear to cause pronounced downstream effects on excitability, neurotransmission and neuronal health [31]. 

Investigations are beginning to utilize microbiome manipulation to modify neuropathology and related deficits. Using an antibiotic-based strategy to modify the microbiota load, we previously found a significant increase in ethanol consumption in a binge-like “Drinking in the Dark” (DID) paradigm when C57BL/6J mice were treated with a non-absorbable antibiotic cocktail (ABX) [34]. Our subsequent study demonstrated that sodium butyrate (SB) supplementation protected against the ABX-induced increase in ethanol intake in mice within the DID model [35]. 

In this study, we set forth to test the hypothesis that SB protects against the ABX-induced increase in ethanol consumption behaviors through the modulation of neuroinflammatory responses. Effects of SB supplementation on the levels of pro- and anti-inflammatory cytokines and changes in glial cells, microglia and astrocytes, were tested in mice in ABX-enhanced ethanol consumption model. 

## 2. Results

### 2.1. ABX Treatment Enhanced Ethanol-Induced Increases in IL-6 and TNF-α mRNA Levels in Mouse Brain

Chronic ethanol intake is shown to induce a neuroinflammatory response that can lead to behavioral alterations [36]. In our previous work, ABX treatment led to enhanced ethanol consumption in C57BL mice [34]. To investigate the potential for ABX to aggravate ethanol-induced neuroinflammation, we tested for mRNA levels of both pro-inflammatory and anti-inflammatory cytokines in whole brains of mice. In agreement with previously published findings, exposure to ethanol during DID induced an increase in mRNA levels of all tested cytokines, such as MCP-1 and IL-1β (Figure 1A–D; H_2_O-DID vs. H_2_O). Further, ABX treatment enhanced the ethanol-induced mRNA levels in pro-inflammatory cytokines TNF-α and IL-6 (Figure 1B,C; ABX-DID vs. H_2_O-DID). Ethanol-induced mRNA levels of pro-inflammatory MCP-1, IL-1β, and anti-inflammatory IL-10 were not altered by ABX treatment (Figure 1A,D,E; ABX-DID vs. H_2_O-DID).

### 2.2. Sodium Butyrate Supplementation Reduced Ethanol-Induced and ABX-Enhanced Increases in mRNA Levels of Key Inflammatory Cytokines 

SB supplementation significantly reduced the mRNA expression levels of both pro-inflammatory (MCP-1, TNF-α, IL-6, IL-1β) and anti-inflammatory (IL-10) cytokines induced by ethanol with or without ABX treatment (Figure 2). Without ABX treatment, SB supplementation reversed the ethanol-induced increases in mRNA levels for MCP-1, IL-1β and IL-10 (Figure 2A,D,E; SB-DID vs H_2_O-DID) and caused reduction in TNF-α and IL-6 levels which however did not reach statistical significance (Figure 2B,C; SB-DID vs H_2_O-DID). When treated concomitantly with ABX, SB supplementation completely reversed the ABX-ethanol-induced increases in the levels of MCP-1, IL-1β and IL-10 and reversed the ABX-aggravated portion of ethanol-induced increases in mRNA levels of TNF-α and IL-6 (Figure 2A–E; ABX+SB-DID vs. ABX-DID). 

### 2.3. Sodium Butyrate Supplementation Prevented Ethanol-Induced and ABX-Enhanced Activation of Hippocampal Microglial Cells 

We next assessed changes in microglia, known as important contributors to neuroinflammation [37]. We used IHC staining of hippocampal tissues for Iba-1, a microglial marker, to clarify the alterations in the hippocampal microglia during different treatments. As expected, ethanol significantly increased Iba-1 positive cell number compared to control mice (Figure 3A,B; H_2_O-DID vs. H_2_O). This effect was significantly potentiated with ABX treatment during the DID procedure (Figure 3A,B; ABX-DID vs. H_2_O-DID). Importantly, SB supplementation dampened ABX-enhanced increase in the number of Iba-1 positive cells, indicating reduction in neuroinflammatory response (Figure 3A,B; ABX+SB-DID vs. ABX-DID).

Parallel to IHC findings, ethanol induced an increase in Iba-1 mRNA level with a statistical trend. This effect was further slightly enhanced with ABX treatment, as consistent with the data on Iba-1-positive cell number (Figure 3C; H_2_O-DID vs. H_2_O and ABX-DID vs. H_2_O-DID). Inversely, SB supplementation dramatically reduced the mRNA level of Iba-1 (Figure 3C, ABX+SB-DID vs. ABX-DID).

With regard to morphological changes, ethanol induced more branched microglia with thicker, numerous and longer extensions and more prominent cell bodies (somas) compared to no ethanol condition (Figure 3A,D,E). ABX treatment with ethanol resulted in somewhat smaller soma and thick but less numerous extensions compared to those parameters for ethanol alone condition (Figure 3A,D,E). Microglia with SB supplementation with ABX and ethanol had a morphology that was intermediate between water and ethanol conditions with less and shorter extensions compared to those in the DID group but longer and more numerous than microglia in the ABX-DID group (Figure 3A,D). Changes were significantly different between the groups on the extension length and soma size but not for number of extensions (Figure 3E). Overall, ethanol induced microglia with typical neuroinflammation profile while ABX enhanced ethanol effects. SB supplementation prevented changes induced by ethanol with and without ABX treatment as shown with the decrease in microglia numbers and reversal of changes in morphological parameters.

### 2.4. Sodium Butyrate Supplementation Prevented Ethanol-Induced and ABX-Enhanced Reduction in the Activity of Hippocampal Astrocytes 

We assessed changes in astrocytes using immunofluorescence staining of GFAP, a selective astrocytic marker. The GFAP-positive cell number showed an opposite trend from the one in microglia. Ethanol lowered GFAP positive cell number (Figure 4A,B; H_2_O-DID vs. H_2_O) and ABX treatment further aggravated this effect (Figure 4A,B; ABX-DID vs. H_2_O-DID). Importantly, SB supplementation protected against the decrease in GFAP-positive cell number induced by ethanol without or with ABX treatment (Figure 4A,B; SB-DID vs. ABX+SB-DID or H_2_O-DID). 

Changes in GFAP intensity showed a similar trend as seen in GFAP-positive cell count with the difference that ethanol without and with ABX treatment caused a similar extent of reduction; this was restored with SB supplementation (Figure 4A,C). In addition, changes in mRNA levels of GFAP showed a pattern similar to IHC findings on GFAP intensity (Figure 4D). Ethanol without and with ABX treatment reduced the mRNA level of astrocyte, even though these decreases did not reach statistical significance (Figure 4D, H_2_O-DID and ABX-DID vs. H_2_O). SB supplementation significantly increased the GFAP mRNA expression (Figure 4D, ABX+SB-DID vs. ABX-DID or H_2_O). 

## 3. Discussion

We previously found that supplementation with SB was able to protect from ABX-enhanced voluntary ethanol consumption in mice in a binge-like drinking model (DID) [34,35]. Our current study focused on the hypotheses that neuroinflammatory response at least partially accounts for the ABX impact on ethanol drinking and that SB protects from this increase through modulation of neuroinflammation. We tested the effects of ABX and SB within the same experimental setting used for the study of drinking behavior in C57BL/6J mice [35]. Our findings suggest that ABX aggravated the neuroinflammatory response induced by ethanol and that SB supplementation prevented this effect. This conclusion is supported by findings of pro- and anti-inflammatory mediators as well as effects on glial cells, microglia and astrocytes. 

Consistent with previous body of literature [24,25] and our own findings [35], ethanol exposure in the DID procedure induced increases in mRNA levels of key pro-inflammatory cytokines, such as TNF-α, MCP-1, IL-6, IL-1β. ABX treatment further enhanced this effect which was notable for TNF-α and IL-6 but not MCP-1 or IL-1β. An important finding of the study was that supplementation with SB prevented ethanol-induced neuroinflammatory response without and with ABX treatment as observed with mRNA levels of all tested pro- and anti-inflammatory cytokines. In our previous study, SB supplementation prevented ABX-induced increase in ethanol consumption but not the baseline drinking levels [35]. Accordingly, when translated into the ethanol drinking behavior, these findings suggest that reduction in essential metabolites such as butyrate, as a result of ABX treatment, might have aggravated the component of ethanol effect on the neuroinflammatory response and associated drinking behavior. 

The different roles of pro-inflammatory cytokines in experimental drinking models have been established, suggesting that activation of neuro-immune pathways can lead to modulation of AUD-related signaling and subsequent behaviors [38,39,40]. For example, it is well known that TLR4-pathway activation occurs during chronic ethanol exposure that may result in NF-kB-mediated increases in TNF-α, IL1β and IL-6, which are involved in modulation of reward neurocircuitry and levels of GABAergic, dopaminergic, glutamatergic neurotransmission [41,42,43]. Studies from IL-6 and IL-1β KO mice suggested that these cytokines may modulate ethanol consumption behaviors, where TNF-α and IL-6 have been assigned a role in chronic ethanol drinking [24,40,44]. Accordingly, the effects of ABX were observed on these cytokines in our study of 4-wk of DID exposure which most likely represents an acute-on-chronic condition. In our study, we also found ethanol-induced increase in anti-inflammatory IL-10 and an enhancement of this effect after treatment with ABX, findings which are consistent with other reports [25,45]. The increase on IL-10 levels mostly occurs as a result to the initial inflammatory response and together with the findings on TNF-α and IL-6 supports the onset of the acute-on-chronic state of inflammation in our model. 

Microglia are primary neuro-immune cells that produce pro-inflammatory cytokines. Acute transformation of microglia into a pro-inflammatory state prepares them to remove damaged neurons [38,40,43,46], yet persistent hyperactivation of microglia can contribute to more neuronal injury [47]. In our study, DID exposure induced an increase in the number of Iba-1-positive microglial cells in hippocampus, which was even further enhanced with ABX-DID treatment, indicating a higher level of neuroinflammation which was also consistent with Iba-1 mRNA expression and changes in the cytokine levels. Furthermore, SB treatment prevented this effect, suggesting a decreased hyperactivation state of microglia. A similar protective effect of butyrate on microglia hyperactivation with a resultant delay in brain aging has been found previously [48].

Morphological shapes of microglia are also indicative of their activation state. Consistent with the existing knowledge, ethanol induced a shape of microglia with enlarged soma and increased bushiness and length of branches, suggesting a state of activation. ABX treatment combined with ethanol resulted in a morphology of microglia resembling a hyperactivated amoeboid shape with less number and shorter branches, whereas morphology of these cells with combination of ethanol, ABX treatment and SB supplementation were intermediate to shapes between control and ethanol conditions. These findings further supported the conclusion that ABX aggravated ethanol-induced microglia hyperactivation and SB supplementation protected from this effect.

Recent studies suggested a key role for astrocytes in the process of neuroinflammation. Astrocytes are a type of glial cells with pleiotropic functions within the brain, including neuronal support, bi-directional interactions with the blood–brain barrier cells and other glial cells [33,49]. Astrocytes have high expression of transporters for glutamate [50] and activated astrocytes were reported to release gliotransmitters including glutamate, D-serine, and ATP [32]. Among number of functions such as regulation of neurotransmitter and water balance and nutritional movements, astrocytes are also known to participate in the inflammatory responses within the brain [33,49]. Changes in the expression of astrocytic marker GFAP and their morphology have been shown in rodent models of alcohol exposure and human brains of long-term alcohol users [51]. Based on that, we evaluated changes in astrocytes with ethanol, ABX and SB treatments. 

Ethanol effects in astrocytes were opposite to those found in microglia. Ethanol induced a significant decrease in the astrocyte cell density and ABX treatment in the presence of ethanol further decreased the number of cells, while SB prevented the decrease caused by combined ABX and ethanol treatment. GFAP intensity paralleled changes seen in cell density with reduction seen with ethanol, however no further decrease was observed with ABX treatment. The mRNA expression changes in GFAP showed a consistent trend with GFAP intensity without reaching level of significance. We also performed a skeleton analysis of astrocyte morphology however these attempts did not result in any meaningful outcomes (data not shown). These analyses are worth future more in-depth investigation of spatiotemporal complexity for glial cells and will need high power, high magnification confocal imaging modality. 

Studies from different animal models of alcohol exposure and human alcoholic brain analyses have demonstrated that GFAP expression can increase or decrease depending on the length of exposure and withdrawal, form of administration, brain regions, species, gender [1]. Thus, DID exposure and ABX treatment may suppress astrocyte activity by lowering the cell density of astrocyte and GFAP expression. This may be a result of increased level of neuroinflammation, favoring reduced neuronal support and modulation of alcohol-induced behaviors. The sustained neuroinflammation is capable of altering membrane expression of neurotransmitter receptors and neurotransmission, involved in reward, and consequently impairing alcohol related behaviors such as spatial learning, cognitive, motor functions [52,53]. Furthermore, the prevention of cell density reduction and GFAP expression in astrocytes with SB supplementation during DID exposure, most probably, originated from the ability of SB to reduce the neuroinflammatory response aggravated by ABX treatment. 

Our findings suggest that the anti-neuroinflammatory effect of SB may at least be partially responsible for its protection against ABX-enhanced ethanol uptake during the DID procedure [1]. To confirm this relationship, we ran a correlation analysis between the ethanol intake data and the neuroinflammatory response, taken as Iba-1 positive cell number. The Pearson correlation analysis demonstrated a positive association between the two parameters (r = 0.35 and p = 0.0017). Number of direct and indirect pathways can be involved in the anti-neuroinflammatory effects of butyrate, including its modulation of vagus nerve, endocrine system, and immune systems (reviewed in [54]). Butyrate serves an energy molecule for enterocytes and is a known regulator of intestinal wall by protecting against “leaky gut” and translocation of bacteria and their components into circulation [11,12,13,14]. Butyrate can act on fatty acid binding G-protein coupled receptors [15], which serve as chemosensors that are capable of combining all the signals from the gut mediators, followed by communicating with the vagus nerve, thus affecting the activities in CNS [10,11,12,48,55]. Another potential pathway is the modulation of a number of CNS neurotransmitters produced by gut microbiota, including glucocorticoids, norepinephrine, and gamma-aminobutyric acid, which are able to cross the blood–brain barrier and take effects in brain potentially via the sympathetic nervous system and hypothalamic-pituitary-adrenal axis [56]. Butyrate has also been extensively studied as a histone deacetylase inhibitor which explains it’s anti-inflammatory effects [14,48]. Future investigation will focus on dissecting the exact pathway(s) involved in butyrate’s protective effects in our experimental model. 

In conclusions, both molecular and cellular findings of the current study supported that butyrate, produced in the intestines by beneficial microbiota species, can modulate the neuroinflammatory responses induced by ABX treatment in the binge-like drinking paradigm, a mechanism which can explain ABX-induced increase in voluntary ethanol consumption. Our future focus will be assessing the exact “footprints” of SB treatment through measurements of activation of signaling pathways and epigenetic changes. These findings further strengthen the importance of the neuroinflammation with respect to alcohol consumption behaviors, allowing for the discovery and development of microbiome-targeted strategies to address alcohol abuse and other substance use disorders.

## 4. Materials and Methods

### 4.1. Animals

Male C57BL/6J mice (Jackson Laboratories, Bar Harbor, ME, USA), 6–8 weeks old, were used in the experiments. Mice were individually housed in a specialized vivarium (automatic humidity, temperature- and light-controlled room) and allowed to acclimate to the environment for at least 2 weeks before being randomly assigned to a specific treatment group. Vivarium-provided rodent chow and drinking bottles were continuously available throughout the whole study, except for when mice had access to ethanol bottles. Mice were housed under a reversed 12 h light/dark cycle (lights on 12:00 am–12:00 pm). Body weight, food consumption, and liquid intake were measured 5 days a week (Monday-Friday) and monitored to ensure mouse health. Measurements were recorded during the light phase, in order to minimize the impact on nocturnal dark phase activities. All animals were treated in accordance with the National Institutes of Health Guide for Care and Use of Laboratory Animals and protocols approved by the USC Institutional Animal Care and Use Committee.

### 4.2. Antibiotic Cocktail and Sodium Butyrate (SB) Treatment

We chose a broad-spectrum antibiotic cocktail treatment (ABX) that leads to efficient depletion and dramatic alterations in the composition of the gut microbiota in adult rodents without entering circulation and directly affecting host tissues due to its lower absorbance [57,58]. The ABX cocktail consisted of 0.5 mg/mL bacitracin (Sigma, St. Louis, MO, USA), 2.0 mg/mL neomycin (GoldBio, St. Louis, MO, USA), and 0.2 mg/mL vancomycin (Thermo Fisher, Waltham, MA, USA). It was provided to mice ad libitum in the drinking water throughout the duration of the study. Anti-fungal pimaricin (1.2 ug/mL, Molekula Group, Irvine, CA, USA) was added to the ABX cocktail, in order to decrease gut fungal overgrowth due to prolonged antibiotic use. SB (Sigma, St. Louis, MO, USA) was provided ad libitum in the drinking water with/without ABX. SB solution of 8 mg/mL in drinking water was freshly prepared every 2 days. SB and ABX+SB solutions were pH matched with the ABX solution to control for acidity effects on daily consumption. 

### 4.3. Drinking in the Dark (DID) Ethanol Self-Administration Procedure

Mice were randomly assigned to 4 treatment groups: H_2_O (n = 9), ABX (n = 9), SB (n = 11) and ABX+SB (n = 11). All mice had 24 h access to their respective drinking bottles for 2 weeks before the start of ethanol experiments. The DID model is widely used to assess differences in binge-like drinking behaviors [59]. We used a modified version of this procedure in our current study where mice had daily limited intermittent access (2 h) to one bottle containing 20% ethanol beginning at 3 h into the circadian dark phase (3:00 p.m. to 5:00 p.m.) 5 days a week (Monday-Friday) for 4 weeks. This approach was shown to maintain consistent ethanol intake levels during the 5 consecutive days of ethanol exposure [3]. During DID, each drinking bottle was exchanged with a designated bottle containing 20% ethanol. 

### 4.4. RT-qPCR

At the end of the DID study, mice were euthanized by CO_2_ asphyxiation and diaphragm puncture. Left hemispheres were excised, flash frozen and stored in −80 °C until further processing. We used 1.5 mL pestle to disrupt and homogenize frozen brain tissue. mRNA was extracted using RNeasy kit (Qiagen, Germantown, MD, USA). RNA concentration was measured using NanoDrop One (Thermo Scientific, Waltham, MA, USA). cDNA was transcribed from 300 ng total RNA using Reverse Transcription system (Promega, Madison, WI, USA) in a final volume of 50 μL. SYBR-Green-based real-time quantitative PCR was performed using the QuantStudio K12 Flex Real-Time PCR (Thermo Scientific, Waltham, MA, USA). PowerUp™ SYBR™ Green Master Mix (Applied Biosystems, Carlsbad, CA, USA) and primers (Integrated DNA Technologies, Coralville, Iowa, USA) were added to a 10 μL reaction system on a 384-well plate. A comparative threshold cycle method was used to calculate expressions relative to control groups. The final results were expressed as fold changes between the sample and the controls corrected with housekeeping gene GAPDH. Primer sequences that were used for the experiments are listed in Table 1.

### 4.5. Immunohistochemistry (IHC)

Right hemispheres of brains after harvesting were post-fixed in 10% Neutral buffered formalin (NBF), cryoprotected in 30% sucrose in phosphate-buffered saline (PBS) for at least 2-3 days and stored in −80 °C until further use. The frozen brain sections containing the hippocampus were mounted in OCT mounting medium (Sakura Finetek, Torrance, CA, USA), and 30 μm-thick sagittal sections were prepared using a Microm HM525 Cryostat (Thermo Fisher, Waltham, MA, USA). Following washes in PBS, sections were incubated in blocking buffer (3% normal serum/0.1% Triton-X/TBS) for 30 min. IHC was performed in free-floating sections with the following primary antibodies: rabbit anti-Iba1 (Catalog # 019-19741; Wako, Richmond, VA, USA) or mouse anti-GFAP (Catalog #3670; Cell Signaling Technology, Danvers, MA, USA). After overnight incubation in primary antibodies, sections were incubated in species appropriate secondary antibodies (both from Thermo Fisher, Waltham, MA, USA): goat anti-rabbit HRP-conjugate (for Iba-1 staining) or goat anti-mouse IgG—Alexa Fluor Plus 550 (for GFAP staining). Sections were mounted onto slides and DAB staining was conducted by using Ultra-Sensitive ABC Rabbit IgG Staining Kit (Thermo Scientific, Waltham, MA, USA) and Metal Enhanced DAB Substrate Kit (Catalog #34065; Thermo Scientific, Waltham, MA, USA). Bright field images were taken by Zeiss Axio Scope A1 microscope (ZEISS, Ontario, CA, USA). Immunofluorescence of sections was imaged in Cytation 5 Cell Imaging Multi-Mode Reader (BioTek, Winooski, VT, USA). 

### 4.6. Image Analysis

DAB (for Iba-1) and fluorescence (for GFAP) images were analyzed using Image J software (National Institute of Health, Bethesda, MD, USA, https://imagej.nih.gov/ij/, 1997–2018) and Photoshop (Adobe Photoshop, Adobe Inc., San Jose, CA, USA). To better present the Iba-1 images, we converted original DAB Iba-1 images to Black&White and threshold images by using Photoshop. Image J cell counter plugin was used to convert all photomicrographs to binary and skeletonized images and to standardize density of microglia and astrocytes, followed by quantification of pixel area per region of interest. Cell counts were measured manually, and analysis was performed in a sample-blinded manner. The following analysis parameters was used: Iba-1-positive cell count/pixels^2^ was calculated by the ratio of Iba-1 positive cell counts to pixel area; GFAP-positive cell count/pixels^2^ was calculated by the ratio of GFAP positive cell counts to pixel area. 

### 4.7. Data Analysis

Graphs and statistical analyses were generated using GraphPad Prism (GraphPad Software Inc., San Diego, CA). Statistical significance of the data was determined using Ordinary one-way ANOVA with Tukey’s multiple comparisons, except for Figure 2C where non-paired Mann–Whitney test was used. Differences were considered significant when the *p* value is less than 0.05. *p* values are indicated as * *p* < 0.05, ** *p* < 0.01 and *** *p* < 0.001. Data are presented as Mean ± SEM (standard error of mean).

## Figures and Tables

**Figure 1 ijms-23-15688-f001:**
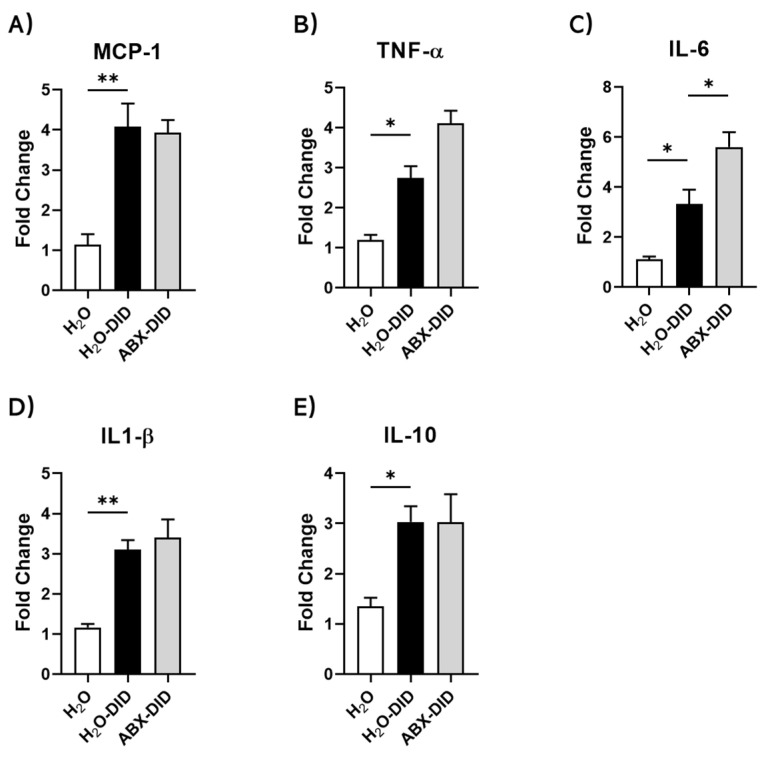
The effects of ABX treatment on mRNA levels of pro- and anti-inflammatory cytokines in mouse brains within the DID model. Ethanol (20%) consumption during the DID procedure induced increases in the mRNA levels of both pro-inflammatory MCP-1, TNF-α, IL-6, IL-1β (**A**–**D**) and anti-inflammatory IL-10 (**E**) cytokines. ABX treatment starting 2 weeks before and continued throughout the DID procedure enhanced mRNA levels of TNF-α and IL-6 and did not affect mRNA levels of MCP-1, IL-1β and IL-10, induced by ethanol. mRNA levels were assessed by real-time RT-qPCR. Data are presented as Mean ± SEM, n = 7–9/group. * *p* < 0.5, ** *p* < 0.01, Ordinary one-way ANOVA with Tukey multiple comparisons.

**Figure 2 ijms-23-15688-f002:**
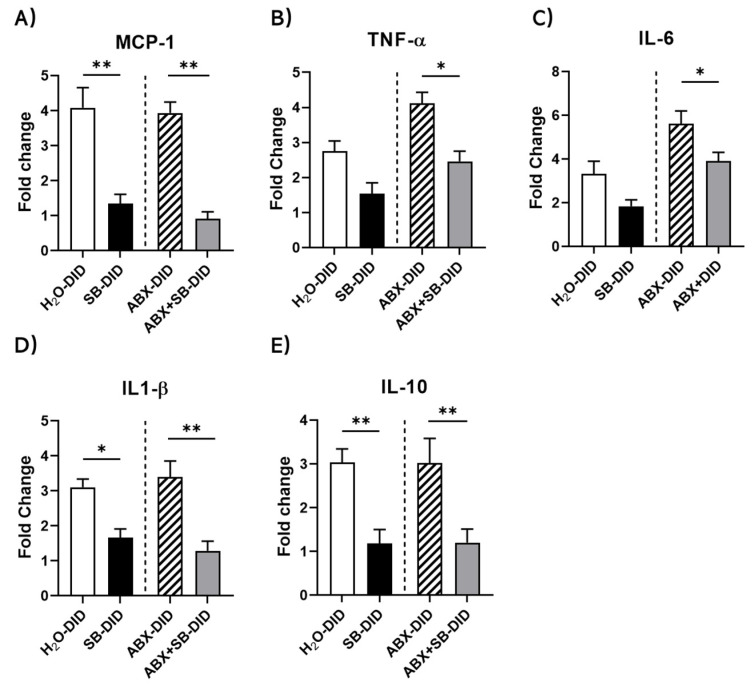
The effects of SB supplementation on DID- and/or ABX-enhanced increases in mRNA levels of pro- and anti-inflammatory cytokines in mouse brains. SB lowered the mRNA expression level of ethanol-induced pro-inflammatory cytokines MCP-1 (**A**), TNF-α (**B**), IL-6 (**C**) and IL-1β (**D**) and anti-inflammatory cytokine IL-10 (**E**) with and without ABX treatment. The mRNA levels were assessed by real-time RT-qPCR. Data are presented as Mean ± SEM, n = 7–9/group. * *p* < 0.5, ** *p* < 0.01, for panels A,B,D,E Ordinary one-way ANOVA with Tukey’s multiple comparisons were used; for panel C, non-paired Mann–Whitney test was used.

**Figure 3 ijms-23-15688-f003:**
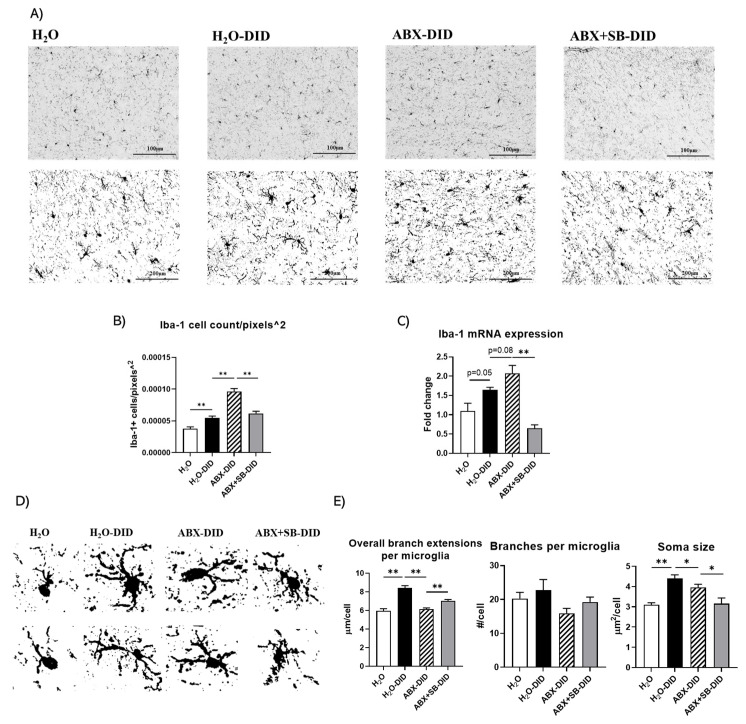
The effects of ABX treatment and/or SB supplementation on DID-induced changes on microglia. (**A**) IHC images of hippocampal microglia. Upper row—20× DAB images (scale bar = 100 μm); lower row—40× threshold images (scale bar = 20 μm). (**B**) Analysis of Iba-1-positive cell number; (**C**) mRNA levels of Iba-1; (**D**) Higher magnification capture of individual microglia to explore their representative morphology; (**E**) Analysis of microglia extension length, extension number and cell body (soma) size. Data are presented as Mean ± SEM, n = 8/group; Ordinary one-way ANOVA with Tukey multiple comparisons, * *p* < 0.05, ** *p* < 0.01.

**Figure 4 ijms-23-15688-f004:**
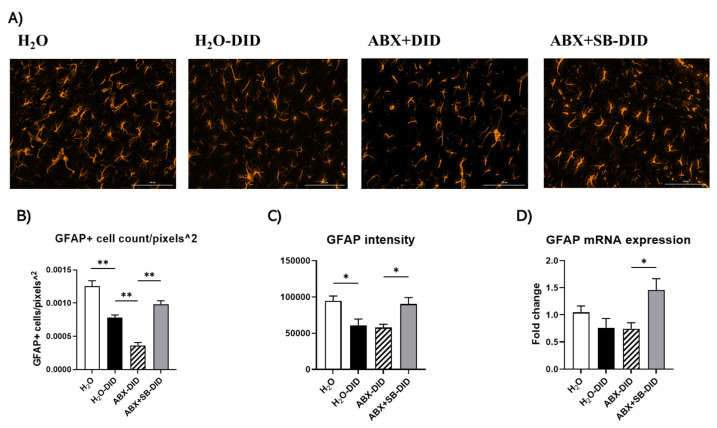
The effects of ABX treatment and/or SB supplementation on DID-induced changes on astrocytes. (**A**) Immunofluorescence images of hippocampal astrocytes; scale bar is 100 μm. (**B**) Analysis of GFAP-positive cell number. The data are shown as GFAP+ cell count/pixels^2^. (**C**) Analysis of GFAP-fluorescence intensity. (**D**) mRNA levels. Data are presented as Mean ± SEM, n = 8/group; Ordinary one-way ANOVA with Tukey multiple comparisons, * *p* < 0.05, ** *p* < 0.01.

**Table 1 ijms-23-15688-t001:** Primer sequences for cytokines tested in RT-qPCR.

Target Gene	Forward Primer (5′ > 3′)	Reverse Primer (5′ > 3′)
MCP-1	GCAGCAGGTGTCCCAAAGAA	ATTTACGGGTCAACTTCACATTCAA
TNF-α	GGTGCCTATGTCTCAGCCTCTT	GCCATAGAACTGATGAGAGGGAG
IL-1β	TGGACCTTCCAGGATGAGGACA	GTTCATCTCGGAGCCTGTAGTG
IL-6	ACAACCACGGCCTTCCCTACT T	CACGATTTCCCAGAGAACATGTG
IL-10	CGGGAAGACAATAACTGCACCC	CGGTTAGCAGTATGTTGTCCAGC
GAPDH	AGGTCGGTGTGAACGGATTTG	TGTAGACCATGTAGTTGAGGTCA

## Data Availability

All data are presented in this paper and can be shared upon request (email: asatryan@usc.edu).

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
