# Peer review of "Sodium Butyrate Supplementation Modulates Neuroinflammatory Response Aggravated by Antibiotic Treatment in a Mouse Model of Binge-like Ethanol Drinking"

_ijms, 2022, doi:10.3390/ijms232415688_

Round 1
Reviewer 1 Report
This is a very interesting submission regarding the role of sodium butyrate in neuroinflamatory response aggravated by antiobiotics in a mouse model of binge-like ethanol drinking. The results are really interesting and the paper has merit. Comments:
-Introduction should be improved by adding text and references for sodium butyrate. This is important and especially if some relevant work exists.
-Statistical analysis: Please add the software
-SB treatment: Have the authors tested other concentrations, as well?
Author Response
We would like to thank the reviewers for providing us some constructive comments. We have addressed those and have a point-by-point answers to each of those below. The revisions are highlighted in yellow throughout the manuscript.
Reviewer 1
This is a very interesting submission regarding the role of sodium butyrate in neuroinflamatory response aggravated by antiobiotics in a mouse model of binge-like ethanol drinking. The results are really interesting and the paper has merit. Comments:
-Introduction should be improved by adding text and references for sodium butyrate. This is important and especially if some relevant work exists.
We would like to thank the reviewer for the insightful comment. We have revised the Intro accordingly and added some references supporting various functions of butyrate, including the anti-inflammation, energy source, histamine deacetylase inhibitor.
-Statistical analysis: Please add the software
We used Graph Prism for the analysis and graphs. This information was added to the Data analyses section of Methods.
-SB treatment: Have the authors tested other concentrations, as well?
We have carefully picked the concentration of sodium butyrate based on previous publications. We have addressed that in our previous paper (Reyes, Gao, Zhang et al, Alcohol, 2021). Our ongoing follow-up study is testing the escalated concentrations of sodium butyrate (20 and 50mg/ml). The findings will be analyzed and presented in a separate manuscript.
Reviewer 2 Report
The manuscripted titled Sodium Butyrate Supplementation Modulates Neuroinflammatory Response Aggravated by Antibiotic Treatment in a Mouse Model of Binge-like Ethanol Drinking needs minor revision before it could be accepted for publication.
1. Figure 3 B, C, D, E are blurred. Please improve resolution.
2. Line 180: insert space between 100 and um.
3. Lines 288 to 304:
4. I think you need to correlate your data here to explain the potential mechanisn of butyrate.
Author Response
We would like to thank the reviewers for providing us some constructive comments. We have addressed those and have a point-by-point answers to each of those below. The revisions are highlighted in yellow throughout the manuscript.
Reviewer 2
The manuscripted titled Sodium Butyrate Supplementation Modulates Neuroinflammatory Response Aggravated by Antibiotic Treatment in a Mouse Model of Binge-like Ethanol Drinking needs minor revision before it could be accepted for publication.
- Figure 3 B, C, D, E are blurred. Please improve resolution.
The figure resolution of Fig 3 has been increased. However, the panels may be appearing blurred because of the pfd conversion of the file. So better resolution may not be guaranteed for viewing, however we will submit separate high-resolution images (if required) to ensure required quality of figures in the final publication.
- Line 180: insert space between 100 and um.
The space was added as advised. In addition, the manuscript was proof-read to improve the English text and correct any other mistakes and typos.
- Lines 288 to 304:
- I think you need to correlate your data here to explain the potential mechanisn of butyrate.
We guessed that items 3 and 4 pertain to the same comment.
Per reviewer’s suggestion, we ran a quick correlation between our alcohol intake data and the Iba1 positive cell number as a marker for neuroinflammation. This approach is justified as the ethanol intake and neuroinflammation are part of the same animal study. We could have used the cytokine profile but would take more time to look into individual cytokines. The analysis shown below demonstrated a good correlation between the neuroinflammation and ethanol intake. We added only the analysis numerical data (r and p values) in the Discussion.
